# CoMO-NAS: Core-Structures-Guided Multi-Objective Neural Architecture Search for Multi-Modal Classification

## ABSTRACT

Most existing NAS-based multi-modal classification (MMC-NAS) methods are optimized using the classification accuracy. They can not simultaneously provide multiple models with diverse preferences such as model complex and classification performance for meeting different users' demands. Combining NAS-MMC with multi-objective optimization is a nature way for this issue. However, the challenge problem of this solution is the high computation cost. For multi-objective optimization, the computing bottleneck is pareto front search. Some higher-quality MMC models (namely core structures, CSs) consisting of high-quality features and fusion operators are easier to identify. We find that CSs have a close relation with the pareto front (PF), i.e., the individuals lying in PF contain the CSs. Based on the finding, we propose an efficient multi-objective neural architecture search for multi-modal classification by applying CSs to guide the PF search (CoMO-NAS). In conclusion, experimental results thoroughly demonstrate the effectiveness of our CoMO-NAS. Compared to state-of-the-art competitors on benchmark multi-modal tasks, we achieve comparable performance with lower model complexity in shorter search time.

## CCS CONCEPTS

• **Computing methodologies** → **Artificial intelligence**.

## KEYWORDS

Multi-modal fusion, Core structures, Neural architecture search, Multi-objective optimization, Classification

## 1 INTRODUCTION

The success of multi-modal fusion architectures owes much to their design [7, 16, 32]. Recently, with the advantages of escaping labor-intensive and challenging architectural design processes, neural architecture search (NAS) [18] has experienced unprecedented interest. NAS has achieved significant success in discovering optimized multi-modal feature fusion strategies, surpassing manually designed methods [14, 15, 24, 39]. However, these approaches focus solely on the need for high-accuracy fusion architectures. In addition to accurate predictions, practical applications also require NAS-MMC methods to find computationally efficient network architectures, such as low power consumption in mobile applications,

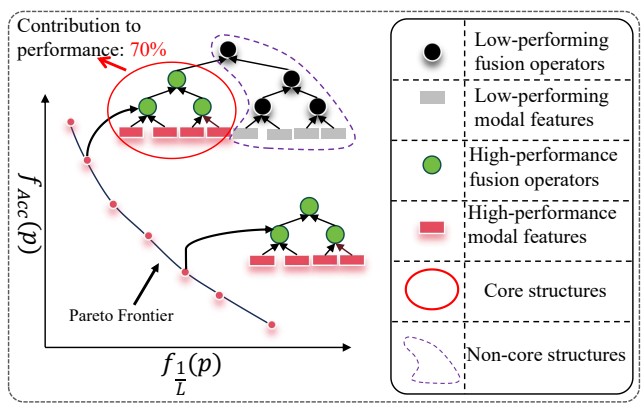

**Figure 1: The relationship between core structures (CSs) and Pareto frontier (PF). The individuals lying on PF contain CSs.**

low latency in autonomous driving applications, and deployability on edge devices like smartphones. It is widely observed that as network architecture complexity increases, predictive performance continues to improve. This has sparked competition between maximizing predictive performance while minimizing complexity, leading to the natural idea of introducing multi-objective optimization.

Among the many diverse MMC-NAS methods, evolutionary algorithms (EAs) have garnered widespread attention due to their population-based nature and flexibility [44]. They offer a viable alternative to traditional ML-oriented approaches, especially within the scope of multi-objective NAS. Generally, EAs involve an iterative process where improvements are gradually made to individuals within the population by applying variations to selected individuals and recombining parts of multiple individuals. Despite being easily scalable to handle multiple objectives, most existing EA-based NAS methods are still single-objective driven. Additionally, a computational bottleneck for utilizing evolutionary algorithms in multi-objective optimization lies in the search for the Pareto frontier, requiring significant computational resources.

To address the aforementioned issues, we propose a method called core structures-guided multi-objective neural architecture search (CoMO-NAS). As illustrated in Figure 1, core structures are substructures composed of high-performing features and fusion operators, which often play a decisive role in the architecture's performance. These core structures align with the optimization objectives of multi-objectives, representing lower-complexity and high-performance substructures themselves. Additionally, through observations of existing advanced MMC-NAS methods, we find that the optimal fusion architectures identified by MMC-NAS typically include some excellent core structures. Building upon these findings, we further explore and discover that core structures frequently appear in the final Pareto frontier, and even more complex solutions

within the Pareto frontier also contain core structures. Therefore, we propose utilizing core structures to guide multi-objective neural architecture search. To obtain core structures, we first divide the features extracted by the backbone network and predefined fusion operators into two parts: one part contains individually high-performing features and fusion operators constituting the core structure search space, while the other part contains lower-performing features and fusion operators constituting the non-core structure search space. In the first stage, we utilize multi-objective evolutionary algorithms to search for core structures within the core structure search space, obtaining a portion of the Pareto frontier. In the second stage, we use the core structures along with the non-core structure search space to search for architectures with higher complexity and better performance, forming a complete Pareto frontier.

Our research has been validated on multiple multi-modal datasets, showcasing optimal performance in terms of efficiency, complexity, and accuracy. Specifically, our contributions include:

- We find that the core structures (CSs) that consist of features and fusion operators with higher performance in NAS-MMC can be used to guide the Pareto front (PF) search in multi-objective optimization because the individuals in PF often contain the CSs. This strategy is able to significantly improve search efficiency and solution quality.

- With above the finding, we propose a method called core structure-guided multi-objective neural architecture search (CoMO-NAS). To the best our knowledge, CoMO-NAS introduces the concept of multi-objective algorithms in the field of MMC-NAS for the first time. It can efficiently provide multiple optimization solutions for different scenarios.

- We conducted extensive experimental comparisons on multiple multi-modal tasks, and the results show that compared to state-of-the-art multi-modal feature fusion methods, CoMO-NAS has significant advantages in terms of search time and the number of model parameters.

## 2 RELATED WORK

### 2.1 Multi-Modal Fusion

Multi-modal fusion networks have demonstrated clear advantages over single-modal networks in various applications such as action recognition and sentiment analysis [9, 17, 33, 35, 43]. However, effectively combining multi-modal features to better utilize information remains a significant challenge [36]. multi-modal fusion techniques are typically divided into two main categories: the first category involves handcrafted fusion based on domain knowledge, such as early fusion, fusion of low-level features, late fusion, and fusion of decision-level features [16]. Some approaches also involve feature fusion at intermediate layers to facilitate later fusion and improve performance, for example, CentralNet [31] and MMTM [30], which connect potential representations from each layer and pass them as auxiliary information to deeper layers. However, this approach significantly increases the parameter count of multi-modal fusion models. Additionally, there are recent works proposing dynamic multi-modal fusion, a novel method that adaptively fuses multi-modal data and generates data-relevant forward paths during inference [7, 19, 23, 34, 46]. The second category involves applying

neural architecture search to automatically find the optimal fusion architecture [15]. Compared to the first category, this approach eliminates the laborious traditional handcrafted design and generally achieves better fusion results, albeit at the expense of requiring substantial computational resources and time.

### 2.2 Multi-Modal Neural Architecture Search

Neural Architecture Search (NAS) [18] has been introduced to automate the design of neural models, aiming to discover efficient architectures with competitive performance. This trend has sparked researchers' interest in migrating NAS to the field of multi-modal fusion, leading to the proposal of a series of multi-modal neural architecture search methods to automatically design optimal fusion network architectures. Perez-Rua et al. first explored and validated the feasibility of using NAS methods to address this issue. They proposed a search framework called MFAS [25], which automatically selects single-modal features from all candidate features as inputs to the fusion module. However, due to MFAS's adoption of the black-box optimization algorithm SMBO, each update step requires training a set of DNNs, resulting in low efficiency. Additionally, MFAS only utilizes concatenation and fully connected (FC) layers for single-modal feature fusion, where the stack of FC layers poses a significant computational burden. Yin et al [39]. introduced a two-layer gradient-based search scheme named BM-NAS, allowing simultaneous search of input features and fusion operations for each multi-modal fusion module. However, it forces cells to have different predecessors, leading to a narrowed search space and potentially suboptimal results. EDF [15] utilizes evolutionary neural architecture search to find multi-modal fusion architectures for chemical structures, achieving superior results. However, due to the inherent limitations of evolutionary algorithms, the time cost is high. To address the time cost issue brought by evolutionary NAS, DC-NAS [14] proposes a divide-and-conquer evolutionary neural architecture method, greatly improving search efficiency.

While the aforementioned methods have achieved significant success in the field of multimodal fusion, they are all driven by the goal of performance optimization. This may lead to a tendency to solely pursue performance during the search process, consequently increasing the complexity of the models. In addition to the high demand for accuracy, practical applications also require NAS-MMC methods to discover network architectures that are computationally efficient, catering to scenarios with low power consumption and memory constraints. To address this issue, we propose a multi-objective multimodal neural architecture search framework guided by core structures. This framework utilizes evolutionary algorithms to compensate for the shortcomings of existing methods.

## 3 METHODS

### 3.1 Definition and Motivation

To avoid confusion, we provide precise definitions for certain terms here. A population consists of individuals, where each individual corresponds to a multi-modal classification (MMC) model encoded in the form of a tree, typically represented as a vector in post-order traversal for ease of displaying experimental results in this paper. All representations extracted from different modalities are collectively referred to as features. The space composed of high-quality features

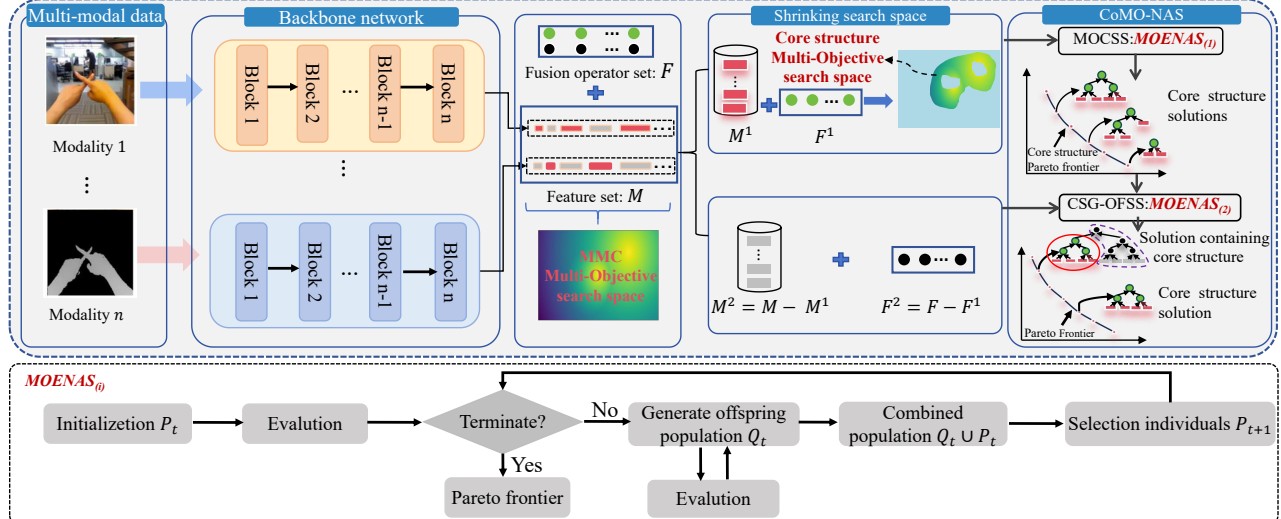

**Figure 2: The framework of CoMO-NAS**

**Table 1: Comparison results of the quality of Pareto solutions between traditional multi-objective algorithms and CoMO-NAS on the CB dataset. $C$ denotes the model complexity. The vector [.] represents a fusion architecture, where the numbers with/without negative signs denote fusion operators and modality features, respectively.**

| $HV[4]\uparrow$ | $ID$ | Pareto frontier solution | $C\downarrow$ | $Acc\uparrow$ (%) |
|---|---|---|---|---|
| | 1 | $[7, 9, 4, 1, -0, -0, -4]$ | 7 | 87.80% |
| | 2 | $[7, 9, 4, 7, 2, 8, 5, -2, -0, -0, -0, -4]$ | 13 | 88.47% |
| Other | 3 | $[8, 7, 2, -0, -4]$ | 5 | 87.53% |
| 52.37 | 4 | $[7, 9, 8, 4, 2, -3, -2, -0, -4, -0]$ | 9 | 88.07% |
| | 5 | $[7, 2, 7, 5, -2, -0, 4, 1, -0, -0, -4]$ | 11 | 88.40% |
| | 6 | $[8, 7, -4]$ | 3 | 86.65% |
| | 1 | $[8, 7, 4, 3, 1, -0, -0, -1, -1]$ | 9 | 88.40% |
| | 2 | $[7, 4, -1]$ | 3 | 86.71% |
| CoMO-NAS | 3 | $[4, 8, 7, 2, -2, -0, -4]$ | 7 | 88.21% |
| 58.27 | 4 | $[8, 7, 4, -1, -1]$ | 5 | 87.53% |
| | 5 | $[8, 7, -4]$ | 3 | 86.71% |
| | 6 | $[8, 7, 4, 3, 1, 5, -3, -0, -0, -1, -1]$ | 11 | 88.57% |

and fusion operators is termed as the core structure search space, while the remaining constitutes the non-core structure search space.

Our work involves two key concepts: the Pareto frontier and core structures, which are closely intertwined. The core structures is a submodule that plays a central role in multi-modal fusion architectures. The performance of most fusion architectures stems from those core structures, typically composed of high-performing modality features and fusion operations, as depicted in Figure 1. Core structures are usually architectures with good performance and low complexity, thus they often appear on the Pareto frontier and both of them exits close relationship. By leveraging core structures, we can identify partial solutions within the Pareto frontier. However, relying solely on core structures may not be sufficient

when considering both good performance and complexity. Nevertheless, since these high-performing and complex architectures often contain core structures, identifying core structures first can guide the search for high-performing and complex architectures, forming a complete Pareto frontier to meet various scenario demands. Additionally, utilizing core structures to guide the search process can significantly accelerate the search for the optimal Pareto frontier by drastically reducing the evaluation of numerous ineffective architectures, thus saving a considerable amount of time.

To provide a more intuitive understanding of the relationship between the two concepts, we conducted experiments on the CB dataset. First, we employed traditional multi-objective evolutionary algorithms. Next, we utilized core structures to guide the search across the entire PF. The specific algorithm steps are outlined in the following sections. The results are presented in Table 1. From the experimental results, we can draw the following conclusions: (1) in terms of the HV [4] indicator, CoMO-NAS discovers a better Pareto frontier compared to traditional multi-objective evolutionary algorithms. Here, a higher HV indicator indicates better performance; (2) We observe that the Pareto frontier of CoMO-NAS includes core structures with IDs 4 and 5, and most other solutions are guided by core structures. For example, IDs 1 and 6.

## 3.2 CoMO-NAS

In this paper, we propose a core structure-guided multi-objective neural architecture search for finding the optimal Pareto front. CoMO-NAS consists of three steps: (1) unimodal feature extraction, (2) multi-objective core structures search (MOCSS) and (3) core structures-guided optimal Pareto frontier search. The main framework of CoMO-NAS is shown in Figure 2.

*3.2.1 Unimodal Feature Extraction.* This study follows previous work on multi-modal fusion, such as MFAS, MMTM, and BM-NAS, using pre-trained single-model neural network models as feature extractors. We extract raw features from the intermediate layers of

these models because neural network architectures typically have a layered or block-like structure, which naturally suits this extraction method. As the feature extractors used for different modalities vary, resulting in significant differences in the dimensions of the raw features—for example, text modalities may yield one-dimensional features, images two-dimensional, and videos three-dimensional—we employ global average pooling to uniformly convert them into feature vectors. This facilitates feature alignment, promotes subsequent feature fusion, and simultaneously reduces computational complexity for ease of processing.

*3.2.2 MOCSS: Multi-Objective Core Structures Search.* To ensure the efficiency and convenience of vectorized feature fusion within the entire architecture, we employ five basic fusion operators for feature fusion. Here, we define $x_i$ as the vector feature, $n$ denotes the number of vector features to be fused, and the superscript indicates the index of the fused vector feature. In this context, the fusion operator set $F$ encompasses the following operations:

(1) *Concatenation*: The information from vector features is fused as follows:

$$o(x_i) = [x_i^1, x_i^2, \cdots, x_i^{|n|}], \tag{1}$$

where $[\cdot, \cdot]$ is the concatenation operator.

Element-wise fusion operators require that the dimensions of input vectors are the same, hence different vector features need to be mapped into the same dimension space by a linear function before fusing. This can be achieved using a fully-connected layer (FC) without any activation function.

(2) *Addition*: The information from vector features is fused as follows:

$$o(x_i) = \text{FC}(x_i^1) + \text{FC}(x_i^2) + \cdots + \text{FC}(x_i^{|n|}). \tag{2}$$

(3) *Multiplication*: The information from vector features is fused as follows:

$$o(x_i) = \text{FC}(x_i^1) \circ \text{FC}(x_i^2) \circ \cdots \circ \text{FC}(x_i^{|n|}), \tag{3}$$

where $\circ$ denotes Hadamard product, namely element-wise multiplication.

(4) *Max*: The information from vector features is fused as follows:

$$o(x_i) = \max(\text{FC}(x_i^1), \text{FC}(x_i^2) \cdots, \text{FC}(x_i^{|n|})), \tag{4}$$

where max is element-wise max, also called max-pooling.

(5) *Average*: The information from vector features is fused as follows:

$$o(x_i) = \frac{1}{|n|}(\text{FC}(x_i^1) + \text{FC}(x_i^2) + \cdots + \text{FC}(x_i^{|n|})), \tag{5}$$

where + denotes element-wise addition, also called average-pooling.

To obtain core structures, we can reduce the entire search space to the core structure search space by evaluating the performance of each feature and fusion operator at relatively low cost. Specifically, given $n$ features represented as $M_1, M_2, ..., M_n$, and a single-modal classifier $f$, we pass each feature $M_i$ to $f$ and select the top $k_1$ features with higher performance to form a high-quality feature set $M^1$. Given $m$ fusion operators represented as $F_1, F_2, ..., F_m$, and a multi-modal classifier $h$, we obtain the classification performance of each fusion operator $F_i$ by replacing the fusion manner of $h$ with $F_i$ and select the top $k_2$ fusion operators with higher performance to form a high-quality fusion operator set $F^1$. The space composed

of $M^1$ and $F^1$ is referred to as the core structure search space. Next, in the core structure search space, we use a multi-objective evolutionary algorithm for searching. After $N1$ iterations of **MOENAS**, we obtain the core structure population and partially obtain the Pareto frontier solutions during the iteration process. Please refer to Section 3.3 for specific details about the **MOENAS**.

*3.2.3 CSG-OPFS: Core structures-guided optimal Pareto frontier search.* The core structure have a close relationship with the final Pareto frontier obtained through multi-objective search. This is because the core structures are characterized by low complexity and high precision. The final Pareto frontier typically includes some core structures, but to obtain a comprehensive optimal Pareto frontier, we need to integrate the remaining features and fusion operators. This is because the remaining features and fusion operators often provide complementary information, enhancing the overall performance of the fusion architecture. In the previous stage, we already obtained the core structures, and some more precise multimodal fusion architectures usually include the core structures as well. Therefore, we can quickly determine a high-quality search subspace by leveraging these core structures, which consists of the neighborhood surrounding the core structure. Here, the neighborhood refers to the continuous addition of substructures composed of the remaining features $M^2$ and the fusion operator set $F^2$ to the core structure, forming a multi-modal fusion architecture. To obtain the final Pareto frontier, we utilize the **MOENAS** algorithm, taking the core structure along with the remaining features $M^2$ and $F^2$ as new inputs. Through evolutionary algorithms, we can adaptively evaluate the fusion architectures around each core structure, achieving the goal of searching the neighborhood and ultimately obtaining the complete Pareto frontier.

## 3.3 Search Strategy

For both stages, we employ the NSGA-II algorithm to search for the core structure and the final Pareto frontier. The overall algorithm is presented in Algorithm 1.

(1) Individual Encoding and Decoding: To cover various fusion strategies more flexibly, each individual in the population is encoded as a binary tree, capable of encompassing any fusion strategy. In this representation, leaf nodes represent modality features, while internal nodes represent fusion operators. Due to the inherent nature of binary trees, when there are k features, there must be k-1 fusion operators. For the decoding process, each individual corresponds to a multi-modal classification model. The binary tree can be decoded into a multi-modal classification model through the following steps: 1) Channeling modality features represented by the leaf nodes of the individual encoding tree into fully connected layers (FC) for feature alignment, facilitating feature fusion, and adding batch normalization (BN) layers to enhance model convergence speed and improve generalization; 2) Conducting feature fusion based on the fusion operators represented by the internal nodes; 3) Directing the fused features through FC and Softmax layers for the final prediction output.

(2) Population Initialization: For the CoMO-NAS framework, there are two search stages, each with a different initialization approach. In the first stage, $K$ non-repeating individuals are generated in a random distribution within the core structure search subspace.

---

**Algorithm 1** The pseudocode of CoMO-NAS

---

**Input**: Fusion operator set $F$, feature set $M$, population size $K$, maximum iterations for core structure search $N_1$, maximum iterations for optimal PF fusion architecture search $N_2$.

1: Core structure search space $CSSS$ by separating high-quality feature set $M^1$ and fusion operator set $F^1$;

2: Non-core structure search space $NCSSS$ is achieved by utilizing the remaining feature set $M^2$ and fusion operator set $F^2$.

3: $P_0 \leftarrow$ Initialize a population comprising $K$ individuals through the $CSSS$;

4: Evaluate the accuracy and complexity of each individual in $P_0$ as fitness values.

5: **for** $i \in [1, N_1]$ **do**

6:    $Q_i \leftarrow$ **Algorithm 2**$(P_i, CSSS)$;

7:    Evaluate the accuracy and complexity of each individual in $Q_i$ as fitness values.

8:    $O_i = P_i \cup Q_i$;

9:    $P_{i+1} \leftarrow$ Get the next generation population from $O_i$ using the non-dominated sorting method in NSGA-II;

10: **end for**

11: Core structure population $P'_0 \leftarrow P_{N_1}$;

12: **for** $j \in [1, N_2]$ **do**

13:    $Q_j \leftarrow$ **Algorithm 2**$(P'_j, NCSSS)$;

14:    Evaluate the accuracy and complexity of each individual in $Q_j$ as fitness values.

15:    $O_j = P'_j \cup Q_j$;

16:    $P'_{j+1} \leftarrow$ Get the next generation population from $O_j$ using the non-dominated sorting method in NSGA-II;

17: **end for**

18: **Return** The optimal Pareto front population

---

Specifically, $K$ unique features are randomly sampled from the feature pool, along with $k - 1$ fusion operators, and randomly combined into binary tree structures. This process is repeated $K$ times to create the initial population $P_0$. In the second stage, the $K$ core structures obtained from the first stage are utilized as the initialization population $P_{N1}$.

(3) Individual Evaluation: In CoMO-NAS, the network architecture and weight parameters corresponding to each individual encoding are first obtained on the training data. Subsequently, the accuracy and complexity of the network are calculated based on the validation data, serving as the two objectives for individual evaluation to facilitate subsequent multi-objective algorithms.

Objective1-Accuracy or Weighted F1 score: The first objective shown in Equations 6 is to maximize accuracy or weighted F1 score. The choice between accuracy and weighted F1 score as the first optimization objective depends on the characteristics of the dataset. When dealing with highly imbalanced data, we use the weighted F1 score for better evaluation, which is consistent with previous methods.

$$\max f_{Acc}(p) = \text{Accuracy}(p), \text{F1-W} = F1_{weighted}(p) \quad (6)$$

Where $Accuracy$ and $F1_{weighted}$ represent the functions computed for the individual $p$.

---

**Algorithm 2** The generation of offspring population

---

**Input**: The current population $P_g$, mutation structure sampling space $MSSS$, crossover rate $r_1$, mutation rate $r_2$.

1: $Q_g \leftarrow \emptyset$;

2: **for** $j \in [1, |P_g/2|]$ **do**

3:    $p_1, p_2 \leftarrow$ Select two mating individuals in $P_g$ using binary tournament selection;

4:    $d_1 \leftarrow$ Randomly generate a number in [0,1];

5:    **if** $d_1 \geq r_1$ **then**

6:       $p'_1, p'_2 \leftarrow$ Generate offspring by applying the crossover operator to parent individuals $p_1$ and $p_2$;

7:    **else**

8:       Directly copy parent individuals $p_1$ and $p_2$ to offspring $p'_1$ and $p'_2$;

9:    **end if**

10:    $d_2 \leftarrow$ Randomly generate a number in [0,1];

11:    **if** $d_2 \geq r_2$ **then**

12:       $p'_1, p'_2 \leftarrow$ Mutate individuals $p'_1$ and $p'_2$ using the mutation structures derived from $MSSS$;

13:    **end if**

14:    $Q_g \leftarrow Q_g \cup \{p'_1, p'_2\}$;

15: **end for**

16: **Return** $Q_g$

---

Ojective2-Complexity: The second objective, as shown in Equation 7, is to minimize the complexity of each individual. Here, complexity refers to the length of the architecture, which is the sum of the number of features and fusion operators. Our goal is to achieve higher performance by using fewer features and fusion operators, while also effectively reducing the parameters and redundancy in the architecture. Existing methods often face an issue of maximizing performance by incorporating as many features and fusion operators as possible, leading to redundancy. The final fusion architectures they search for are often extensive, with a portion of them potentially yielding only marginal benefits. MoCo-NAS partially mitigates this problem by reducing the complexity of the model.

$$\min f_L(p) = \text{Features}(p) + \text{Fusion operators}(p) \quad (7)$$

(4) Evolutionary Strategy: Using a reproduction method based on crossover and mutation, offspring populations are generated, as outlined in Algorithm 2. Initially, two pairing individuals are selected from the current population $P_g$ through binary tournament selection. Subsequently, reproduction involves duplication, crossover, and mutation operations to yield offspring individuals. The entire algorithm comprises two stages: the first stage involves searching for core structures and partial Pareto frontiers, while the second stage utilizes core structures to explore the neighborhoods around them, aiming to achieve the complete optimal Pareto front. Different crossover and mutation rates are applied for distinct objectives. During the core structure search, following traditional evolutionary algorithm principles, the probability of crossover operation significantly surpasses that of mutation. The objective here is to explore core structures, where mutation generates a substructure from the core structure search space, substituting a specific substructure within the individual. Conversely, for exploring the neighborhoods around core structures to attain the entire Pareto

front, opposite crossover and mutation rates are adopted. Utilizing an extremely low crossover rate and very high mutation rate aims at exploring the neighborhoods around core structures, resulting in a high probability of mutation. Mutation generates a substructure from the non-core structure space and incorporates it into the individual to explore the neighborhoods around core structures. The low crossover rate aims to prevent disruption of the core structures themselves.

## 4 EXPERIMENTS

### 4.1 Experimental Settings

In our experiments, all methods are implemented using TensorFlow 2.0.3. Our computational environment consistes of Ubuntu 16.04.4 with 16GB GPU memory, 512GB DDR4 RDIMM, 2X 40-Core Intel Xeon CPU E5-2698 v4 @ 2.20GHz, and NVIDIA Tesla P100. It is worth noting that the GPU configuration used in this paper is the same as the MFAS, EDF, and DC-NAS architectures.

(1) Parameter settings: a) Training of DNNs: All deep neural network models are trained using the Adam algorithm. The learning rate is set to 0.001, with a first-moment exponential decay rate of 0.9 and a second-moment exponential decay rate of 0.999. Each network undergoes training for 100 epochs. To prevent overfitting, if the performance of a multi-modal neural network model does not improve after 10 epochs, the training process will be halted. b) CoMO-NAS: To efficiently utilize GPU resources, the population size is set to a multiple of the number of GPUs. We employed seven NVIDIA Tesla P100 GPUs for the CB dataset, NUS dataset, NTU RGB-D dataset, and EgoGesture dataset , with a population size of 28. The number of iterations is set to 10, with 6 iterations for core structure search and 4 iterations for searching the local region of core structures. During the core structure search phase, the crossover rate is 0.9, and the mutation rate is 0.2. During the search for the local region of core structures, the crossover rate is 0.1, and the mutation rate is 0.8. Considering that the MM-IMDB dataset is relatively simpler compared to the first two tasks, we used four NVIDIA Tesla P100 GPUs with a population size of 20. The rest of the settings are consistent with the above datasets.

(2) Evaluation metrics: We utilize accuracy as the evaluation metric on CB, NUS, NTU RGB-D, and EgoGesture datasets, where higher values indicate better performance. On the MM-IMDB dataset, we employ F1-W as the evaluation metric, also aiming for higher values. Additionally, we use the Hypervolume (HV) to measure the quality of the final Pareto front obtained by the CoMO-NAS algorithm, validating the effectiveness of the core-guided search method compared to traditional approaches. It is important to note that larger HV values correspond to better algorithm performance.

### 4.2 Multi-Modal Datasets

We validated five popular multi-modal datasets: (1) ChemBook-10k (CB) [15] dataset, designed for chemical structure image recognition in patent retrieval studies, which contains 100,000 chemical structure images distributed into 10,000 categories. (2) NUS-WIDE-128 (NUS) [29] dataset, which contains 43,800 images divided into 128 categories. We chose a subset of 10 categories totalling 23,438 images from this dataset. (3) MM-IMDB [1] dataset for the multi-label film genre classification task, which contains a total of 23 categories.

**Table 2: The accuracy on the CB and NUS dataset are reported**

| Method | CB | NUS |
|---|---|---|
| Advanced fusion operators | | |
| MBL | 82.38±0.32 | 70.60±0.29 |
| MFB | 87.94±0.32 | 71.34±0.40 |
| TFN | 73.45±0.30 | 63.66±1.22 |
| LMF | 82.81±0.18 | 71.74±0.70 |
| PTP | 85.08±0.11 | 71.83±0.50 |
| Multi-modal methods | | |
| TMC (ICLR21) | 77.88±0.20 | 72.73±0.30 |
| TMOA (AAAI22) | 86.81±0.09 | 72.60±0.48 |
| EmbraceNet | 85.85±0.09 | 72.43±0.38 |
| AWDR | 86.66±0.16 | 72.44±0.66 |
| RAMC | 85.36±0.46 | 72.51±0.67 |
| EDF (TEVC2021) | 88.33±0.29 | 74.18±0.70 |
| DC-NAS (AAAI24) | 88.50±0.32 | 74.20±0.32 |
| CoMO-NAS | **88.69±0.38** | **74.24±0.29** |

The dataset is divided into a training set of 15,552 films, a validation set of 2,608 films, and a test set of 7,799 films. (4) NTU RGB-D [26] dataset for multi-modal action recognition task containing 60 categories. The training, validation and test sets include 23,760, 2,519 and 16,558 samples, respectively. (5) EgoGesture [45] dataset for multi-modal gesture recognition task containing 83 categories. The training set of this dataset includes 14,416 samples, the validation set includes 4,768 samples, and the test set includes 4,977 samples.

### 4.3 Comparison Methods

To validate the effectiveness and efficiency of the proposed algorithm, we selected several state-of-the-art algorithms and compared them with CoMO-NAS. These peer competitors can be broadly categorized based on whether the architecture is manually designed. The first category is MMC whose fusion architectures are designed by human experts, including MBL [11], MFB [40], TFN [42], LMF [21], PTP [8], TMC [7], TMOA [20], AWDR [37], RAMC [10], Maxout MLP [5] , VGG Transfer [28], Two-stream [27], GMU [1], CentralNet [31], Inflated ResNet-50 [2], Co-occurrence [13], MMTM [30], VGG-16 + LSTM [38], C3D + LSTM + RSTTM [22], I3D [3], ResNext-101 [12], and MTUT [6]. The second category is NAS-based MMC methods including EDF [15], MFAS [25], BM-NAS [39], 3D-CDC-NAS2 [41], and DC-NAS [14].

### 4.4 Performance Comparison

**Results on CB and NUS**. To reduce the influence of variability stemming from data partitioning and network initialization, we partitioned each dataset uniformly into training and testing subsets. More precisely, instances from every category were randomly distributed, allocating 80% for training and 20% for testing. All methodologies were evaluated under identical data partitioning conditions. The experiments were iterated five times for each approach using consistent configurations, and the average performance, alongside standard deviation, was reported.

According to the aforementioned algorithm, we first select high-quality features and fusion operations to form the core structure

**Table 3: Multi-label genre classification results on MM-IMDB dataset. Weighted F1 (F1-W) is reported.**

| Method | Modality | F1-W(%) |
|---|---|---|
| Unimodal methods | | |
| Maxout MLP (ICML13) | Text | 57.54 |
| VGG Transfer (ICLR15) | Image | 49.21 |
| Multi-modal methods | | |
| Two-stream (NIPS14) | Image + Text | 60.81 |
| GMU (ICLR17) | Image + Text | 61.70 |
| CentralNet (ECCV18) | Image + Text | 62.23 |
| MFAS (CVPR19) | Image + Text | 62.50 |
| BM-NAS (AAAI22) | Image + Text | 62.92±0.03 |
| DC-NAS (AAAI24) | Image + Text | 63.70±0.11 |
| CoMO-NAS (ours) | Image + Text | **63.84±0.16** |

**Table 4: Action recognition results on NTU RGB-D dataset**

| Method | Modality | Acc (%) |
|---|---|---|
| Unimodal methods | | |
| Inflated ResNet-50 (CVPR18) | Video | 83.91 |
| Co-occurence (IJCAI18) | Pose | 85.24 |
| Multi-modal methods | | |
| Two-stream (NIPS14) | Video + Pose | 88.60 |
| GMU (ICLR17) | Video + Pose | 85.80 |
| MMTM (CVPR20) | Video + Pose | 88.92 |
| CentralNet (ECCV18) | Video + Pose | 89.36 |
| MFAS (CVPR19) | Video + Pose | 89.50±0.60 |
| BM-NAS (AAAI22) | Video + Pose | 90.48±0.24 |
| DC-NAS (AAAI24) | Video + Pose | 90.88±0.07 |
| CoMO-NAS (ours) | Video + Pose | **90.94±0.02** |

**Table 5: Gesture recognition results on EgoGesture dataset**

| Method | Modality | Acc (%) |
|---|---|---|
| Unimodal methods | | |
| VGG-16 + LSTM (NIPS14) | RGB | 74.70 |
| C3D + LSTM + RSTTM | RGB | 89.30 |
| I3D (CVPR17) | RGB | 90.33 |
| ResNext-101 (FG19) | RGB | 93.75 |
| VGG-16 + LSTM (CVPR14) | Depth | 77.70 |
| C3D + LSTM + RSTTM | Depth | 90.60 |
| I3D (CVPR17) | Depth | 89.47 |
| ResNeXt-101 (FG19) | Depth | 94.03 |
| Multi-modal methods | | |
| VGG-16 + LSTM (CVPR17) | RGB + Depth | 81.40 |
| C3D + LSTM + RSTTM | RGB + Depth | 92.20 |
| I3D (CVPR17) | RGB + Depth | 92.78 |
| MMTM (CVPR20) | RGB + Depth | 93.51 |
| MTUT (3DV19) | RGB + Depth | 93.87 |
| 3D-CDC-NAS2 (TIP21) | RGB + Depth | 94.38 |
| BM-NAS (AAAI22) | RGB + Depth | 94.96±0.07 |
| DC-NAS (AAAI24) | RGB + Depth | 95.22±0.05 |
| CoMO-NAS (ours) | RGB + Depth | **95.25±0.03** |

search space. For example, for the CB dataset, we choose features $M_1$, $M_3$, $M_4$, $M_7$, $M_8$, and fusion operations $F_1$, $F_2$, $F_5$. For the NUS dataset, we select features $M_2$, $M_4$, $M_6$, and fusion operations $F_1$, $F_2$, $F_5$. Subsequently, we search for core structures and utilize local algorithms based on these core structures to explore the entire Pareto frontier. To comprehensively showcase the advancements of MMC-NAS, we followed the experimental settings of EDF. We compared MMC-NAS with some advanced multi-modal fusion operators and existing sophisticated multi-modal fusion methods. From the results in Table 2, it's evident that, compared to advanced fusion operators, we achieved a significant lead by employing basic fusion operators along with our search strategy. Among multi-modal methods, except for EDF and DC-NAS [14], all others are non-NAS methods. Clearly, the performance of MMC-NAS surpasses manual selection. By utilizing core structures to guide the search across the entire Pareto frontier, even when balancing model complexity and accuracy, we can achieve performance comparable to state-of-the-art single-objective methods like EDF and DC-NAS [14].

**Results on MM-IMDB**. To ensure fair comparison with other explicitly multimodal fusion approaches, we adopted the same neural network backbone models as BM-NAS and DC-NAS to extract various modality features, using weighted F1 score as the evaluation metric. The specific parameter settings are as follows: population

size $N$ is 20, population iterations $T$ is 10, fusion vector dimension $FD$ is 128, and modality features are reusable simultaneously. As shown in Table 3, CoMO-NAS achieves performance comparable to the current state-of-the-art architectures compared to existing multimodal classification methods.

**Results on NTU**. To ensure the fairness of the experimental results, we followed the data preprocessing pipelines of BM-NAS and DC-NAS. Specifically, we used Inflated ResNet-50 [2] and Co-occurence [13] as feature extractors for the skeleton and video modalities. For the CoMO-NAS evolutionary algorithm parameters, due to our approach of searching the Pareto frontier from the perspective of core structures, which narrows down the search space, the required population size and number of iterations are smaller than those of traditional evolutionary algorithms. For instance, while state-of-the-art DC-NAS may require 15 generations of population, we only need 10 generations to discover a Pareto frontier solution that matches the performance of DC-NAS. We set the population size to 28, the number of iterations to 10, the fusion modality dimension to 64, and allowed for the reuse of modality features. In Table 4, our method exceeds most baseline methods while achieving comparable performance with the state-of-the-art DC-NAS, ensuring both model complexity and performance objectives.

**Results on Ego**. We followed the methods of BM-NAS and DC-NAS, using ResNeXt-101 [12] as the backbone network for RGB and depth video modalities. CoMO-NAS was compared with various single-modal and multi-modal methods in terms of performance. The experimental settings for CoMO-NAS included a population size of 28, 15 iterations, reusable modality features, and a fusion dimension of 32. The experimental results on the EgoGesture dataset are presented in Table 5. Compared to other unimodal/multimodal methods, CoMO-NAS achieved fusion performance comparable to the state-of-the-art method DC-NAS.

**Table 6: Comparison of complexity, model parameters, time (GPU hours) and classification performance (CP) of generalized multi-modal NAS methods.**

| Method | Dataset | Complexity | Parameters | Time | CP (%) |
|--------|---------|-----------|-----------|------|--------|
| EDF | NUS | 27 | 0.65M | 12.08 | 74.18 |
| DC-NAS | NUS | 17 | 0.53M | 4.61 | 74.20 |
| CoMO-NAS | NUS | **9** | **0.29M** | **2.19** | **74.24** |
| EDF | CB | 31 | 4.41M | 126.84 | 88.33 |
| DC-NAS | CB | 19 | 3.06M | 87.56 | 88.50 |
| CoMO-NAS | CB | **11** | **2.66M** | **38.15** | **88.69** |
| BM-NAS | MM-IMDB | 11 | 0.65M | 1.24 | 62.94 |
| DC-NAS | MM-IMDB | 5 | 0.42M | 1.19 | 63.70 |
| CoMO-NAS | MM-IMDB | **5** | **0.42M** | **0.68** | **63.84** |
| MMTM | NTU | - | 8.61M | - | 88.92 |
| MFAS | NTU | 12 | 2.16M | 603.64 | 89.50 |
| BM-NAS | NTU | 14 | 0.98M | 53.68 | 90.48 |
| DC-NAS | NTU | 27 | 0.92M | 24.94 | 90.88 |
| CoMO-NAS | NTU | **9** | **0.42M** | **8.86** | **90.94** |
| BM-NAS | Ego | 16 | 0.61M | 20.67 | 94.96 |
| DC-NAS | Ego | 15 | 0.39M | 7.30 | 95.22 |
| CoMO-NAS | Ego | **7** | **0.26M** | **3.53** | **95.25** |

**Table 7: Ablation study of CoMO-NAS**

| Version | MOCSS | CSG-OPFS | Time | Complexity | Acc (%) |
|---------|-------|----------|------|-----------|---------|
| CoMO-NAS$_1$ | False | False | 62.70 | 13 | 88.47 |
| CoMO-NAS$_2$ | True | False | 58.98 | 13 | 88.37 |
| CoMO-NAS | True | True | **38.15** | **11** | **88.67** |

## 4.5 Search Efficiency Comparison

This section aims to compare CoMO-NAS with several powerful MMC baseline methods, including MFAS, EDF, BM-NAS, DC-NAS, and MMTM [30], focusing on search efficiency, complexity, model size, and performance to demonstrate its advanced capabilities. The research results have been comprehensively summarized in Table 6. From the table, it can be observed that on five complex datasets, CoMO-NAS is able to find architectures with comparable performance but lower complexity and smaller model size compared to other methods, while leading in efficiency. For example, on the NUS and CB datasets, our efficiency is nearly four times that of the EDF method and twice that of DC-NAS, while the model complexity is halved. On the NTU RGB-D and EgoGesture datasets, while achieving comparable performance, we gain a significant advantage in model complexity. The time consumption for searching the optimal fusion model is reduced by almost six times compared to the latest method of BM-NAS, and by half compared to DC-NAS. This is attributed to our core structure-guided multi-objective neural architecture search framework, which significantly narrows down the search space, effectively avoids evaluating a large number of poorly performing models, and imposes multi-objective constraints on model complexity, thereby greatly reducing model redundancy and accelerating search speed.

**Table 8: Ablation study of CoMO-NAS**

| Strategy | fusion strategy | Complexity | Acc (%) |
|----------|----------------|-----------|---------|
| CoMO-NAS | [7,8,7,1,-4,4,3,-4,-4,-4,-0] | 11 | **88.67** |
| Random1 | [9,8,7,2,5,4,-2,-2,-2,-2,-0] | 11 | 87.90 |
| Random2 | [9,4,7,2,8,1,-4,-4,-4,-0,-1] | 11 | 88.20 |
| Random3 | [7,1,0,4,-4,-4,9,8,-4,-0,-1] | 11 | 88.20 |
| Random4 | [3,2,7,-4,4,1,0,-0,-4,-4,-1] | 11 | 87.39 |

## 4.6 Ablation Study

To provide a more in-depth analysis of the proposed CoMO-NAS, we conducted a detailed examination of each component and hyperparameter of CoMO-NAS via the ablation experiments on the CB dataset which is the most complex among the five datasets.

**Analysis of the Impact of MOCSS and CSG-OPFS Stages on CoMO-NAS**: To further investigate the impact of MOCSS and CSG-OPFS on CoMO-NAS, we conducted a comprehensive analysis of three scenarios of CoMO-NAS. According to the results in Table 7, we draw the following conclusions: compared to searching the entire space, utilizing core structures to guide multi-objective neural architecture search can lead to finding architectures with lower complexity but comparable performance in a shorter time. For example, the time was reduced from 62.70 hours to 38.15 hours, shortening the duration by 24.55 hours. From the results of CoMO-NAS$_2$ compared to CoMO-NAS, when only using the core structure search space without expanding the surrounding local space, the overall performance of the searched architectures declined. This is because low-quality features and fusion operators within the local space can also complement each other, thereby enhancing the overall performance of the architecture.

**Analysis of Core Structure Selection Strategies**: To investigate the impact of core structure selection on subsequent Pareto frontier exploration, four experiments were conducted. In the first experiment, high-quality features and fusion operations were employed to search for core structures, while the subsequent four experiments involved the random selection of features and fusion operations for core structure exploration. The experimental results presented in the Table 8 unequivocally demonstrate that employing high-quality features and fusion operations for core structure search leads to significantly superior outcomes compared to randomly selecting features and fusion operations.

## 5 CONCLUSION

In this paper, we has proposed a multi-objective neural architecture search method guided by core structures to address the limitations of existing MMC-NAS methods, which had focused solely on achieving high performance while ignoring the varying demands of different applications for classification performance and model size. Furthermore, it has resolved the issue of model redundancy that has arisen from pursuing high performance in existing MMC-NAS methods. By establishing the relationship between core structures and the Pareto frontier and utilizing core structures to guide the search across the entire Pareto frontier, the method has avoided evaluating numerous ineffective architectures, thereby significantly improving search efficiency. Extensive experiments has validated the advantages of CoMO-NAS.

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
