# OpenReview forum: "CoMO-NAS: Core-Structures-Guided Multi-Objective Neural Architecture Search for Multi-Modal Classification"
_acmmm.org/ACMMM/2024/Conference — MM2024 Poster_

### Official Review · Reviewer_g2n4 · 2024-05-23

**Rating:** 2
**Confidence:** 3

**Summary:**

Combining NAS-MMC with multi-objective optimization faces the challenge of high computation cost. The authors propose a multi-objective neural architecture search for multi-modal classification by applying constraint satisfaction (CS) to guide the Pareto front (PF) search (CoMO-NAS). The authors claim that the proposed model has lower complexity and shorter search time.

**Strengths:**

The problem addressed in this paper is of significant value, and the authors' approach is quite innovative.

**Limitations:**

1. The abstract and contribution sections of the paper need further enhancement. The current presentation makes it difficult to highlight the main contributions of the paper. The authors should clearly and explicitly state the key differences between the proposed method and the existing state-of-the-art methods to emphasize the paper's contributions.
2. The METHODS section needs reorganization, as the current presentation is somewhat difficult to follow. The authors introduce a new architecture, but they need to clearly explain why this architecture can reduce the existing computational cost. This explanation should include theoretical analysis as well as experimental validation and analysis.
3. The steps of the proposed method are numerous, but many steps are not clearly explained. For instance, in section 3.2.2, the feature fusion process involves five basic fusion operators. The authors should explain the purpose and rationale behind using these operators. Similarly, many other steps need to be elucidated to make the paper more coherent, centered around the main theme, and more readable.

**Suitability:**

2

---

### Official Review · Reviewer_D7Q6 · 2024-05-24

**Rating:** 5
**Confidence:** 3

**Summary:**

The article discusses the limitations of existing neural architecture search (NAS)-based multi-modal classification (MMC-NAS) methods, pointing out that they typically optimize only for classification accuracy and cannot simultaneously provide diverse models that meet different user demands, such as model complexity and classification performance. Combining NAS-MMC with multi-objective optimization is a natural solution to this issue, but the main challenge lies in the high computational cost, particularly in the Pareto front (PF) search process.
The study finds that some high-quality MMC models, known as core structures (CSs), which consist of high-quality features and fusion operators, are easier to identify. These core structures have a close relationship with the Pareto front, meaning that individuals on the Pareto front contain the core structures. Based on this finding, the authors propose an efficient multi-objective neural architecture search method for multi-modal classification (CoMO-NAS), which uses core structures to guide the Pareto front search. Experimental results thoroughly demonstrate the effectiveness of CoMO-NAS. Compared to state-of-the-art competitors on benchmark multi-modal tasks, this method achieves comparable performance with lower model complexity and shorter search time.

**Strengths:**

I believe this article demonstrates significant novelty, as it introduces the concept of multi-objective optimization algorithms into the field of Multimodal Classification Neural Architecture Search (MMC-NAS) for the first time. By guiding Pareto Front (PF) searches through Core Structures (CSs), a new multi-objective neural architecture search method (CoMO-NAS) is proposed. In terms of technical correctness, the article, through observation and experimentation, reveals the important role of core structures in high-performance multimodal fusion architectures and elucidates the relationship between these structures and the Pareto Front. The CoMO-NAS method improves search efficiency and solution quality by incorporating core structures of high-performance features and fusion operators into the multi-objective multimodal neural architecture search process. To validate the correctness of the CoMO-NAS method, the article presents extensive experimental comparisons across multiple multimodal tasks, demonstrating CoMO-NAS's superior performance in terms of efficiency, complexity, and accuracy. The experimental results indicate that compared to state-of-the-art multimodal feature fusion methods, CoMO-NAS offers significant advantages in search time and the number of model parameters.

**Limitations:**

The article does not discuss the limitations of the proposed method or suggest directions for future work. Should we consider extending the CoMO-NAS architecture presented in this paper to other fields beyond multimodal classification tasks in the future?

There are some minor issues in the paper, such as in Table 2 where "CoMO-NAS" is missing "(ours)" and should be consistent with the rest of the table.

**Suitability:**

2

---

### Official Review · Reviewer_B7vF · 2024-05-24

**Rating:** 4
**Confidence:** 2

**Summary:**

The paper titled "CoMO-NAS: Core-Structures-Guided Multi-Objective Neural Architecture Search for Multi-Modal Classification" presents a novel approach to Neural Architecture Search (NAS) that is specifically tailored for multi-modal classification tasks. The authors introduce a method that combines NAS with multi-objective optimization to address the limitations of existing NAS methods, which often focus solely on classification accuracy while neglecting other important factors such as model complexity and computational efficiency. The key innovation of CoMO-NAS is the use of high-quality substructures, termed "core structures" (CSs), to guide the search for the Pareto front, which represents a trade-off between model performance and complexity. The authors claim that their method achieves comparable performance with lower model complexity and in a shorter search time compared to state-of-the-art methods.

**Strengths:**

Novelty: The paper introduces a new concept of using core structures to guide the search in multi-objective optimization for NAS, which is an innovative approach to balancing the trade-off between model performance and complexity.

Theoretical Approach: The theoretical foundation for using core structures to approximate the Pareto front is well-reasoned and aligns with the optimization objectives of multi-objective problems.

Technical Correctness: The methodology is technically sound, with a clear explanation of how core structures are identified and used to guide the search process.

Adequate Evaluation: The authors have conducted extensive experiments on multiple multi-modal datasets, which demonstrates the effectiveness of CoMO-NAS in terms of efficiency, complexity, and accuracy.

Clarity: The paper is well-written and clearly explains the motivation, methodology, and results. The use of figures and tables enhances the reader's understanding of the proposed method.

Applications: The paper discusses the practical applications of NAS in various domains such as mobile applications, autonomous driving, and edge devices, which highlights the relevance of the research.

**Limitations:**

Generalization of Core Structures: The paper does not sufficiently address how the identified core structures generalize across different datasets or modalities. It is crucial to understand if these core structures are dataset-specific or if they have broader applicability.

Exploration of Search Space: While the core structure-guided search is presented as efficient, it may potentially limit the exploration of the entire search space. There is a risk that the method might miss out on other optimal or near-optimal architectures that do not contain the identified core structures.

Sensitivity Analysis: The paper lacks a sensitivity analysis of the hyperparameters used in the CoMO-NAS algorithm. Understanding how changes in these parameters affect the outcome is essential for users to effectively apply the method.

Documentation and Code Availability: The paper does not mention whether the code will be made publicly available. Open-source code would greatly enhance the paper's impact by allowing the community to build upon and further refine the proposed method.

Specificity of Methodology to Multi-Modal Classification: While the paper focuses on multi-modal classification, it is not clear how well the proposed method could be adapted to other types of tasks, such as regression or clustering.

**Suitability:**

2

---

### Official Review · Reviewer_xkT8 · 2024-05-26

**Rating:** 5
**Confidence:** 3

**Summary:**

This paper proposes a core structure-guided multi-objective neural architecture search method based on a interesting finding to efficiently provide multiple optimization solutions for different scenarios. Sufficient experimental evaluations are provided.

**Strengths:**

1.The paper summarized a comprehensive review of the current progress of related work and pointed out the limitations.
2.The method has achieved significant improvements in experiments.
3.This paper is well organized and written.

**Limitations:**

Some concerns and suggestions to improve this work are as follows:
1.While the methodology is interesting, the description of the framework lacks clarity and Figure 2 would benefit from explanatory symbols such as arrows. Please revise to improve reader understanding and ensure clear presentation of methodological processes.
2.The author conducted extensive experimental comparisons on multiple multi-modal tasks. Could you please give some qualitative analysis to explain this?
3.It is recommended that the authors check any potential citation errors and enhance the manuscript's overall quality.

**Suitability:**

2

---

### Meta-Review · Area_Chair_NG3t · 2024-07-01

**Recommendation:** Accept (Poster)
**Confidence:** 3

**Metareview:**

The paper aims to tackle the problem that most existing NAS-based multi-modal classification (MMC-NAS) methods are optimized using the classification accuracy so it can not meet different users' demands. Based on the finding that Core Structures have a close relation with the pareto front (PF), they propose an efficient multi-objective neural architecture search for multi-modal classification by applying CSs to guide the PF search (CoMO-NAS). Experimental results thoroughly demonstrate the effectiveness of our CoMO-NAS. Compared to SOTA benchmarks, the proposed method achieve comparable performance with lower model complexity in shorter search time.

The paper is well-written and easy to understand. The motivation is clear and the proposed method contains signifiant novelty. The reviewers post some questions on the paper writing and numeric problem. The rebuttal has addressed almost all the problem.

One reviewer g2n4 concerns the significant of the novelty while the reviewer address the novelty at the initial reviewer comments. After checking the paper itself, I suggest that the paper is based on interesting observation and the method performs well in both efficient and metrics. And the method will be public which will contribute to the field a lot.

Above all, I suggest to accept this paper.